# Changes in Default Mode Network Connectivity in Resting-State fMRI in People with Mild Dementia Receiving Cognitive Stimulation Therapy

**DOI:** 10.3390/brainsci11091137

**Published:** 2021-08-27

**Authors:** Tianyin Liu, Aimee Spector, Daniel C. Mograbi, Gary Cheung, Gloria H. Y. Wong

**Affiliations:** 1Department of Social Work and Social Administration, The University of Hong Kong, Pok Fu Lam, Hong Kong; tianyin@hku.hk; 2Department of Clinical, Educational & Health Psychology, University College London, London WC1E 6BT, UK; a.spector@ucl.ac.uk; 3Department of Psychology, Pontifícia Universidade Católica do Rio de Janeiro, Rio de Janeiro 22541-041, Brazil; danielmograbi@puc-rio.br; 4Institute of Psychiatry, Psychology & Neuroscience, King’s College London, London SE5 8AF, UK; 5Department of Psychological Medicine, The University of Auckland, Auckland 1010, New Zealand; g.cheung@auckland.ac.nz; 6Sau Po Centre on Ageing, The University of Hong Kong, Pok Fu Lam, Hong Kong

**Keywords:** DMN, dementia, Cognitive Stimulation Therapy (CST), resting-state fMRI, cognitive reserve

## Abstract

Group cognitive stimulation therapy (CST) is a 7-week activity-based non-pharmacological intervention for people with mild to moderate dementia. Despite consistent evidence of clinical efficacy, the cognitive and brain mechanisms of CST remain unclear. Theoretically, group CST as a person-centred approach may work through promoting social interaction and personhood, executive function, and language use, especially in people with higher brain/cognitive reserve. To explore these putative mechanisms, structural MRI and resting-state functional MRI data were collected from 16 people with mild dementia before and after receiving CST, and in 13 dementia controls who received treatment as usual (TAU). Voxel-based morphometry (VBM) and resting-state functional connectivity (rs-FC) analyses were performed. Compared with TAU, the CST group maintained the total brain volume/total intracranial volume (TBV/TICV) ratio. Increased rs-FC in the default mode network (DMN) in the posterior cingulate cortex and bilateral parietal cortices nodes was observed in the CST over TAU groups between pre- and post-intervention timepoints. We provided preliminary evidence that CST maintains/enhances brain reserve both structurally and functionally. Considering the role of DMN in episodic memory retrieval and mental self-representation, preservation of personhood may be an important mechanism of CST for further investigation.

## 1. Introduction

Dementia, including Alzheimer’s disease, vascular dementia, frontotemporal dementia, Lewy body disease, mixed dementia, and other subtypes, affects 50 million people worldwide [1]. Before a cure for dementia can be identified, interventions to maximise functioning and quality of life are necessary [2]. Apart from symptomatic pharmacological treatments, non-pharmacological interventions for dementia are being actively researched to contain this rapidly increasing global disease burden due to population ageing.

Group-based cognitive stimulation therapy (CST) is currently the only manualised non-pharmacological intervention recommended in clinical guidelines for people with mild to moderate dementia to promote cognition and quality of life [3], in view of its effectiveness in delaying cognitive decline and improving quality of life with evidence from meta-analyses, with an effect size for cognitive outcome comparable to anti-dementia medication treatment [4,5]. The standard protocol of group CST involve 14 sessions delivered two times per week for seven weeks with an emphasis on information processing and social functioning [6]. Based on the biopsychosocial model [7], CST emphasises person-centred care, with principles including using reminiscence as an aid to the here-and-now, stimulating language, and stimulating executive function delivered through a range of activities such as sharing on one’s childhood, word games, and categorising objects [6].

Despite existing evidence and potential to further enhance its cognitive benefits, the exact mechanism of CST remains unclear. In an earlier study analysing changes in cognitive domains using the cognitive subscale of the Alzheimer’s Disease Assessment Scale (ADAS-Cog), enhanced language through new semantic links created in CST has been suggested [8]. Another review paper of 12 studies examining the efficacy of CST also suggested that language was the domain most likely to change, owing to the nature of the activities and to the general structure of the CST sessions, for instance, activities designed to stimulate verbal skills and creative language use [9]. Both studies, however, were inconclusive as to the mechanisms. Because of the ecological and complex nature of CST, working out its mechanisms without information on brain-level activities has been challenging.

Brain network connectivity analysis offers an opportunity to advance our understanding of the mechanisms of evidence-based non-pharmacological interventions such as CST. Activity-dependent plasticity refers to the reshaping of brain network connectivity due to behaviorally relevant experience [10]; topologically complex or globally distributed brain networks could undergo reconfiguration during learning [11]. Considering the nature of activities involved in CST, brain networks that support self-representation, personhood, episodic memory, executive function and language may be implicated. These networks of interest include the default mode network (DMN), the central executive network (CEN), and the language network (LAN).

DMN has been suggested to be responsible for internal mentation (e.g., manipulation of episodic and autobiographical memory for self-referential thinking and social function) [12,13,14], and cognition and memory in general [15]. Commonly identified nodes of DMN include the anterior cingulate cortex (ACC), medial prefrontal cortex (MPFC), posterior cingulate cortex (PCC), the precuneus, retrosplenial cortex, and the bilateral parietal lobule [16]. These are among the brain regions first implicated in mild cognitive impairment and Alzheimer’s disease [12,15,17]. DMN activities appear to be regulated by the CEN [18], a network anchored in the dorsolateral prefrontal cortex (dlPFC) and posterior parietal cortex (PPC) [19,20], which supports high-level cognitive functions such as attention control and working memory. In people with dementia, the CEN may also be recruited to support semantic and episodic memory tasks [21], as a compensatory approach [22]. Depending on the subprocesses, language processing can involve a network that include the anterior temporal lobes, dorsomedial PFC, and PCC [23], while a core LAN network anchored in the anterior insula (AI), superior temporal gyrus (sTG), inferior frontal gyrus (iFG), and middle temporal gyrus (mTG) [24]. With increasing age, connectivity of the language network would increase during language tasks, but decrease during resting state [25].

Initial evidence also suggests that such learning may be linked with a person’s cognitive reserve. In a recent study using resting state near-infrared spectroscopy (rsfNIRS) in healthy older people [26], a link between intellectual engagement and cognitive function has been observed in people with high efficiency and clustering in complex network (higher small-worldness), which suggests a role in cognitive reserve. An earlier study, also in healthy older people, found increased connectivity in DMN, CEN, and salience network after exposing to multi-domain activities such as problem-solving, handicraft making [27]. Another study in people with mild dementia receiving CST has shown, using voxel-based morphometry (VBM) analysis, an association between higher normalized grey matter (GM) and white matter (WM) volume at baseline and improvements in general cognition as measured by ADAS-Cog after receiving CST [28]. 

This study aims to explore changes in the DMN, CEN, LAN during resting state associated as putative brain networks implicated in CST, in the context of individual differences in baseline and change in cognitive reserve among people with mild dementia. 

## 2. Materials and Methods

Data were drawn from an MRI substudy of a prospective non-randomised controlled trial comparing participants receiving CST versus treatment as usual (TAU) in Hong Kong Chinese people with mild dementia for cognitive and quality of life outcomes. The study was approved by the Human Research Ethics Committee of The University of Hong Kong (HREC Reference Number: EA1501019). Written informed consent was obtained from both the participants and their carers prior to their participation in the study.

### 2.1. Participants

Participants of the MRI substudy were recruited from a CST trial [4] where they received referrals from community dementia services in Hong Kong. Following inclusion criteria in a key CST trial [4], the inclusion criteria for this substudy were: (1) a clinical diagnosis of dementia; (2) a Cantonese Mini-Mental State Examination (MMSE) score of over 18 [29]; (3) absence of other psychiatric disorders; (4) able to communicate and understand communication; and (5) able to see and hear well enough to participate in a meaningful assessment. The following criteria were added for this MRI study: (6) right-handedness assessed using the Edinburgh Handedness Inventory [30]; (7) no contraindications to MRI, including the presence of cardiac pacemakers, incompatible metal implants, and claustrophobia; and (8) without a history of stroke, heart attack, or head trauma leading to loss of consciousness. A total of 20 participants from the CST group and 17 participants in the control group of the trial were enrolled into this MRI study. Among the 20 participants enrolled in the CST group, follow-up MRI data were available for 16 participants (80% retention rate), with four lost to follow-up or withdrawn. Among the 17 recruited as controls, 15 were eligible for and completed MRI at baseline, and 13 (87% retention rate) completed the follow-up assessments.

### 2.2. Assessments and Procedures

Participants referred from community dementia services were invited for a screening assessment to determine eligibility. After screening and baseline (T0) assessment, participants allocated to the CST group then received a standard group CST protocol with 14 sessions delivered twice a week, of approximately 45 min duration and [6] tailored for Hong Kong Chinese [31]. Participants in the control group received TAU. The same set of assessments were repeated at follow-up (T1), which is upon completion of all CST sessions for the intervention group. 

Behavioural assessments included cognitive performance, mood, communication, as well as the participants’ demographics and proxies for cognitive reserve (years of education and work). Cognitive performance was measured using the ADAS-Cog [32], which covers multiple cognitive domains and has three subscales: (1) memory and learning; (2) language; and (3) praxis. A higher score on ADAS-Cog (score range, 0–70) denotes worse performance. We defined a change in cognition by subtracting T1 scores from T0 score. The Cornell Scale of Depression in Dementia (CSDD) [33] was used to assess participants’ mood based on informant and patient interviews. A higher score suggests worse depression (score range, 0–38). We used the Holden Communication Scale (HCS) [34] to assess participants’ communication, and a higher score indicates more difficulties (score range, 0–48). MRI scans were arranged on average 31 days (SD = 25) after the behavioural assessments at T0, and on average within 42 days (SD = 37) at T1. The average interval between T0 and T1 scan was 145 days (SD = 72).

### 2.3. Magnetic Resonance Imaging (MRI) Data Acquisition and Preprocessing

Structural and functional brain imaging data were acquired using a 3.0-Tesla Philips Achieva whole-body MRI scanner (Philips Healthcare, Best, The Netherlands). Following the echo-planar imaging (EPI) sequence, high-resolution T1-weighted magnetisation-prepared rapid gradient-echo (MP-RAGE) imaging images were acquired for structural morphometric analysis and for preprocessing of fMRI data. Using a 2D gradient EPI sequence, we acquired 180 T2-weighted functional image volumes, using the following parameters: repetition time (TR) = 2000 ms, echo time (TE) = 30 s, flip angle = 90°, number of slices (Nslices) = 32, matrix = 64 × 64, slice thickness = 4 mm, field of view (FoV) = 240 × 240 mm^2^, ascending interleaved slice ordering. Head motion was restricted using firm padding that surrounded the participants’ head throughout the whole scanning process. For resting-state functional imaging, participants were asked to rest inside the scanner with their eyes open, and fixate on a cross in the middle of a projected image inside the scanner, and avoid engaging in any specific thoughts.

T1-weighted MP-RAGE images were preprocessed following the CAT12 protocol (Structural Brain Mapping group, Jena University Hospital, Jena, Germany) implemented as a toolbox in Statistical Parametric Mapping (SPM) 12 package (Institute of Neurology, London, UK). Images were corrected for bias-field inhomogeneities, then spatially normalised and segmented into GM, WM, and cerebrospinal fluid (CSF) within the same generative model [35]. A quality check was performed after the preprocessing, and according to the automated quality insurance protocol of CAT12, all scans included in this study passed the check. All GM images were then smoothed with an 8 mm, full width-at-half-maximum, isotropic Gaussian kernel to minimise between subject cortical variation of the gyrus. 

Functional images were preprocessed using the SPM 12 package. None of the participants were excluded based on the criteria of a displacement > 3 mm in x, y, or z or an angular rotation > 1.5° in any direction. Imaging data were then slice-time corrected and realigned. Functional volumes were co-registered and resliced to a voxel size of 2 mm^3^, normalized to the Montreal Neurological Institute (MNI) template brain, and smoothed with an 8-mm, full width-at-half-maximum, isotropic Gaussian kernel. The denoising process was carried out with the Functional Connectivity toolbox (CONN, V16.a) [36] implemented in the MATLAB environment and included three steps: (1) Linear regression of potential confounding effects in the BOLD signal. To identify the factors, the anatomical component-based noise correction procedure (aCompCor) was used to identify noise components associated with segmented WM and CSF [37], estimated subject-motion parameters [38], and identified scrubbing parameters [39] without affecting intrinsic functional connectivity [40]. (2) Application of a 0.008–0.09 Hz bandpass filter to reduce the effects of low-frequency drift and high-frequency noise [41]. (3) Linear detrending.

### 2.4. Data Analyses

Behavioural data were analysed using SPSS (version 23.0). Independent sample *t* or χ^2^ tests were used to explore differences in characteristics between CST and control groups. Repeated measure ANOVA was used to analyse the areas of significance change from T0 to T1 between the two groups in ADAS-Cog, CSDD, and HCS.

Structural images were analysed using the VBM technique using CAT12, which provides comprehensive information of brain morphometric features while avoiding biases due to structural differences [42]. Total intracranial volume (TICV) was calculated as the sum of the GM, WM, and CSF volumes; and the total brain volume (TBV) as the sum of GM and WM, and the TBV/TICV ratio was then calculated to allow for more direct comparisons between participants. To detect the TBV/TICV ratio changes in participants, paired sample *t*-tests were conducted on the processed images. Repeated-measures ANCOVA and mixed-design ANCOVA with age and gender as nuisance covariates were used to explore within-subject changes between baseline and follow-up conditions.

Resting-state functional connectivity (rs-FC) was performed by applying a seed-based approach using the CONN toolbox [36]. Previously defined regions of interest (ROIs) based on literature were used as seeds for seed-based rs-FC analyses [43]. All seed ROIs were 10 mm diameter spheres specified in MNI coordinates. The ROIs for DMN were the MPFC (−1, 49, 15), PCC (−6, −52, 40), and left and right parietal cortices (lLP and rLP: −46, −70, 36/46, −70, 36). The ROIs for CEN were the left dlPFC (−48, 42, 20) and right dlPFC (48, 42, −2), and left PPC (−24, −58, 68) and right PPC (30, −54, 70) [20]. The ROIs for Language Network were AI (39, 19, 14), sTG (−55, −18, 8), iFG (46, 36, 18), and mTG (62, −37, −3) [24]. See Appendix A for a brain surface map with the different ROIs corresponding to the three networks of interest. Seed-to-voxel analysis was used, in the first-level (individual) analysis, correlations were generated in the CONN toolbox by extracting the mean resting-state BOLD time course from each seed ROI and calculating correlation coefficients with the BOLD timecourse of each voxel throughout the whole brain, and the resulting coefficients were Z transformed [15]. The rs-FC of each network was calculated by averaging the rs-FC of selected seeds within the network, paired sample *t*-tests were conducted to analyse individual level changes. In the second-level between group analysis, the differences of rs-FC of DMN, CEN, and LAN between CST and TAU groups pre- and post- intervention were measured by applying mixed ANCOVA with age, gender, and education as covariates. Pearson’s correlation coefficient was performed to explore bivariate correlations between demographics, cognitive/brain reserve proxies and ADAS-Cog change.

## 3. Results

### 3.1. Demographics, Cognitive Reserve Proxies, and Neuropsychological Profile

Table 1 shows the baseline and follow-up characteristics of the sample with complete baseline and follow-up data. Both groups were characterised by a female dominance with an average age of around 80 years. The CST group had significantly fewer people with any formal education, although both groups had a low level of education, which is common among the current cohort of older persons in Hong Kong due to historical reasons. The TAU group had worse communication as shown in the HCS score. See Appendix A for a correlation matrix of the key variables.

### 3.2. Within-Subject Changes

Paired-sample *t*-test and mixed-design ANCOVA were used to explore within-subject changes. Table 2 summarises the changes and results from paired sample *t*-tests in ADAS-Cog, CSDD, HCS, brain volumes and rs-FC of DMN, CEN, and LAN.

Paired-sample *t*-tests revealed that in the CST group, the TBV/TICV ratio was maintained (*p* = 0.35), yet had significant reduction in the TAU group (*t*(12)= −4.39, *p* < 0.01). The rs-FC of DMN increased in the CST group (*t*(15) = 3.31, *p* < 0.05) but not in the TAU group (*p* = 0.47). The rs-FC of CEN and LAN had no change in CST or TAU group (all *p* > 0.05).

For both groups, repeated measures ANCOVA with age and gender as covariates on TBV/TICV, rs-FC of DMN, CEN, and LAN revealed no significant change over time. Mixed ANCOVA between two groups with age and gender as covariates showed significant interaction effect between timepoints and group on TBV/TICV (*F*(1,25) = 4.36, *p* < 0.05), a marginal interaction between timepoints and group on rs-FC of DMN (*F*(1,25) = 4.25, *p* = 0.05); but no such interaction effect on rs-FC of CEN or LAN. 

Two-sided contrast maps without Bonferroni corrected are shown in Figure 1. Increased correlation in the CST over TAU in the follow-up over baseline was shown in the lLP, rLP, and PCC (see Figure 1).

### 3.3. Predictors of Cognitive Changes

We investigated the bivariate correlation between age, cognitive reserve proxies (education, work), baseline brain reserve proxy (TBV/TICV, rs-FC), and ADAS-Cogs (baseline, follow-up, and change). For the CST group, while age was related to ADAS-Cog score at baseline and follow-up (*r* = 0.56 and 0.47, respectively, *p* < 0.01), it was unrelated to the change in ADAS-Cog score. Years of education in this low-education sample was not related to ADAS-Cog score at baseline or follow-up, or its change. Gender was not related to any ADAS-Cog score either, but female gender was positively correlated with higher baseline TBV/TICV ratio (*r* = 0.56, *p* < 0.05). We observed a significant positive correlation between years of work and improvement in ADAS-Cog (*r* = 0.45, *p* < 0.05), which was not related to baseline or follow-up ADAS-Cog score. Baseline TBV/TICV was not related to baseline ADAS-Cog, but was correlated with follow-up ADAS-Cog (*r* = −0.50, *p* < 0.05), and also positively correlated with improvements in ADAS-Cog score in the CST group (*r* = 0.57, *p* < 0.01). Partial correlation controlling for age and gender revealed a strong correlation between years of work and improvement in ADAS-Cog (*r* = 0.75, *p* < 0.01; Table 3), and the partial correlation between baseline TBV/TICV and improvement in cognition was non-significant (*p* = 0.14). In the TAU group, a significant correlation was observed between baseline TBV/TICV and cognitive improvement (*r* = 0.63, *p* < 0.05), years of work was not correlated with cognitive change. Partial correlation controlling for age and gender still revealed significant correlation between baseline TBV/TICV and improvement in ADAS-Cog (*r* = 0.66, *p* < 0.05). Figure 2a illustrates the relationship between z-transformed years of work and ADAS-Cog change score after controlling for age and gender in both groups; and Figure 2b illustrates the relationship between z-transformed baseline TBV/TICV with standardised ADAS-Cog change score.

Baseline rs-FC in DMN, CEN, or LAN was not related to ADAS-Cog (baseline, follow-up, and change) in all participants in general, except that baseline CEN connectivity was positively correlated with improvements in ADAS-Cog in control group (*r* = 0.68, *p* < 0.05; Appendix A). Partial correlations controlling for age and gender were performed between baseline network connectivity and cognitive improvement in both groups, and no significant results were found (Table 3). To further compare differences in correlations between two groups, Fisher’s Z transformation was performed [44], and comparison of transformed Z revealed marginal difference in the correlation between years of work and improvement in cognition (z = 1.83, *p* = 0.06).

## 4. Discussion

This is the first study exploring the putative impact of CST on functional connectivity of three key brain networks in people with mild dementia, as compared with controls receiving TAU. The results suggested that CST may increase resting-state DMN connectivity, particularly in the PCC, LLP, and RLP nodes, which happened despite a reduction in grey matter volume regardless of the intervention or TAU. Baseline TBV/TICV ratio positively predicted cognitive change in both groups, and years of work is predictive of cognitive improvement only in the CST group. These initial findings provided several directions for future research. 

The potential implication of CST on DMN, but not CEN or LAN, opens an avenue for further research into the mechanisms of action in evidence-based non-pharmacological interventions for dementia. To date, there have been very few studies examining the effects of non-pharmacological intervention for dementia on structural or functional brain changes of the clients. A study among healthy older adults found that a multi-domain combined training reorganized the functional connectivity of DMN by increased rs-FC in PCC regions [45]. A recent study used magnetoencephalography (MEG) to examine changes in brain activity of people with dementia (PWD) before and after receiving non-pharmacological treatment, results revealed a reduced alpha activity in the right temporal lobe and fusiform gyrus, as well as an increased low-gamma activity in the right angular gyrus [46]. Both studies confirm that with dementia or not, aging brains have the capacity for plasticity. The estimated number needed to treat (NNT) with the standard protocol for CST to deliver clinically important cognitive improvement ranged from 6 to 14, suggesting modest efficacy similar to pharmacological intervention [6,47]. Due to the complex nature of the intervention, to further improve the clinical efficacy and reduce the NNT, an understanding of the ‘active ingredient’ in CST as opposed to other services/interventions is needed. The finding of increased resting-state DMN connectivity after CST therefore provided direction for further investigation and refinement of intervention design, with reference to our current understanding of the function of DMN.

In a recent review consolidating previous hypotheses and evidence, DMN is identified as a network responsible for ‘sense-making’ in a social world, by integrating incoming external information with a person’s idiosyncratic memory and knowledge [48]. Research has increasingly shown that, in contrast to the previous task-unrelated, ‘mind wandering’ hypothesis, DMN activities may support ongoing cognition for detailed experiences in a task [49], automated adaptive responses to environmental demands that govern daily lives [50], forming context-dependent models of the situation for establishing shared meaning. This updated understanding of the function of DMN is in line with the emphasis on person-centredness in CST [6] and biopsychosocial theoretical underpinning [7]. Person-centred care emphasises ways to enable a person in making the fullest use of one’s abilities and to remain a social being [51], which is realised in CST principles such as ‘maximising potential, ‘building/strengthening relationships’, ‘respect’, ‘involvement’, and ‘inclusion’. These principles distinguish CST from other interventions that place more emphasis on training and rehabilitation of individual cognitive domains. Future study investigating DMN connectivity change in non-pharmacological interventions with varying level of person-centredness would provide further insight.

The increased connectivity in the medial and bilateral parietal cortices suggested a role of the representation of mental self. Earlier research showed that the medial parietal region may be a nodal structure in self-representation [52] and the lateral parietal cortex may support episodic memory [53]. The role of self in memory has recently been revisited in dementia [54], which points toward the impairment in a sense of self-continuity across temporal contexts linking to a reduced capacity for expressing episodic memory and future thinking. In CST, the principles of ‘using reminiscence as an aid to the here-and-now’ and ‘continuity and consistency between sessions’ may facilitate or reinstate this sense of self-continuity. In a recent study investigating the role of self-referential thinking and memory in people with amnestic mild cognitive impairment [55], both autobiographic recall and narrative conditions were shown to benefit memory, and the authors concluded that the association of information to the ‘self’ in people with cognitive impairment provides a useful schema, which relies on the autonoetic experience of inward reflection and subjective experience of relating to one’s knowledge and experience. Research including behavioural measurements of the subjective experience of self-representation would allow investigation of this putative active ingredient in CST.

The increased DMN connectivity in the CST group happened against a context of a significant reduction in grey matter volume (observed in both groups), which suggests that CST does not alter disease progression in terms of structural neuropathology. In the large-scale prevention research Finnish Geriatric Intervention Study to Prevent Cognitive Impairment and Disability (FINGER) involving 2 years of intensive multimodal intervention for older persons, similar negative findings have been reported on MRI changes in brain volumes, cortical thickness, and white matter lesion volume, despite improved cognition with the intervention [56]. Although it is less likely that cognitively stimulating activities act through structural (versus functional) brain changes, our initial finding of reduced TBV/TICV ratio only in the control group may also suggest it as a potential outcome measure on top of other absolute structural brain changes.

The FINGER study, nevertheless, identified a higher baseline cortical thickness in Alzheimer’s disease signature areas, and possibly a higher hippocampal volume, as predictors for cognitive benefits from the multimodal intervention [56]. On the other hand, the earlier report in healthy older persons using rsfNIRS [26] found that, among people with high mid-life occupation achievement (a proxy for cognitive reserve), those with high small-worldness (associated with higher IQs [15], also a cognitive reserve proxy) are less likely to benefit from intellectual engagement activities. These earlier findings suggested the relevance of baseline brain structural and functional characteristics as predictors of intervention response in healthy older persons. Our results provided support to the use of baseline work years as predictors of response to CST in people with mild dementia, supporting the argument that people with a higher cognitive reserve (i.e., those with excess disability given the same disease severity level) are more likely to benefit, although resting-state functional connectivity of the three brain networks of interest in this study did not predict cognitive gain from CST. On the other hand, baseline TBV/TICV predicted improved cognition in both groups, while baseline CEN connectivity is associated with cognitive improvement in the control group only. The mechanisms by which these brain reserves interact with exposure to everyday activities in people with dementia regardless of non-pharmacological intervention may be of interest for developing daily life strategies to enhance cognition.

This study has several limitations. First, the small sample size limited the possibility in analysis, including a limited statistical power to detect a significant difference in cognitive change between the two groups. Although the study was not designed to detect the effectiveness of CST given the available evidence, this limitation posed difficulties in group comparison and results interpretation, with a lack of support of a behavioural outcome difference between groups. The finding of a difference in DMN connectivity change only in the CST group nevertheless provided a basis for future resting-state fMRI study with sufficient power to detect cognitive outcome changes. Second, due to the exploratory nature of the study, a quasi-experimental design was used, with potential bias in group characteristics. At baseline, the control group had worse communication, while fewer people in the CST group had any formal education. These differences might have contributed to the outcomes and our findings should be interpreted with this consideration. Finally, due to logistics difficulties, the time intervals between behavioural assessment and MRI scan at both the baseline and follow-up were long (CST group: 182 days, SD = 73; control group: 94 days, SD = 26), which might have introduced noise and uncertainties in the data. Notwithstanding these limitations, this study provided evidence for the first time of a brain functional change in people with mild dementia after exposure to a non-pharmacological intervention recommended by clinical guidelines, and suggested directions for further research that may improve its efficacy, as well as a proposed mechanism of action that would be relevant to a wide range of person-centred interventions for dementia. 

## 5. Conclusions

This is the first study to explore the neuropsychological mechanisms of CST using resting-state functional connectivity MRI. We provided preliminary evidence that CST maintains/enhances brain reserve both structurally and functionally. The role of DMN in episodic memory retrieval and mental self representation suggested further investigation into the preservation of personhood as an important mechanism of CST and other person-centred intervention strategies.

## Figures and Tables

**Figure 1 brainsci-11-01137-f001:**
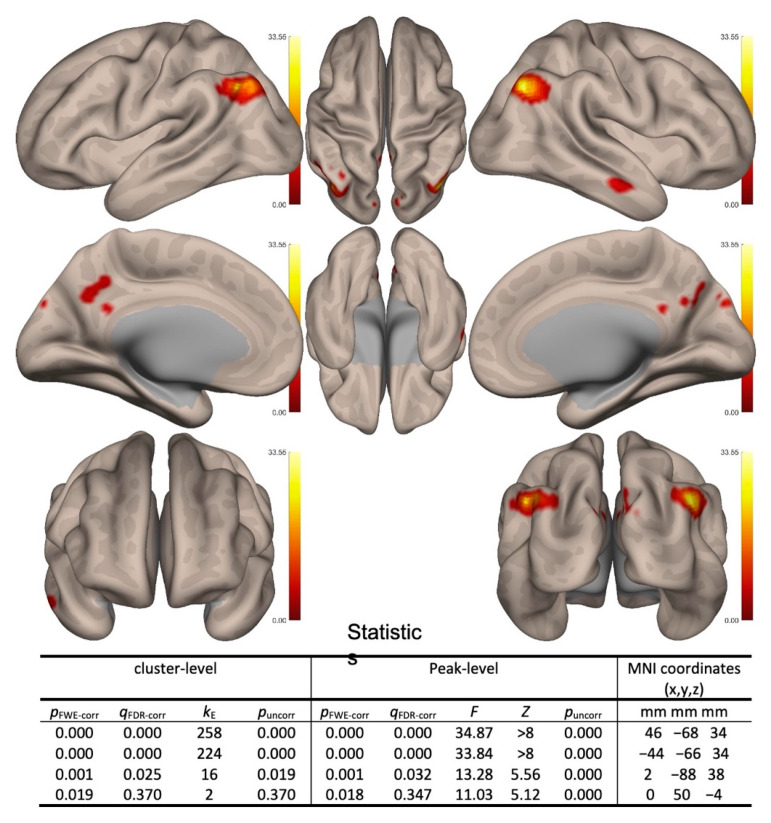
Between conditions (follow-up vs. baseline) and between subjects (cognitive stimulation therapy (CST) vs. treatment as usual (TAU)) second-level contrast maps with significant results. Positive contrasts (follow-up > baseline and CST > TAU) are shown in red colors. (Note: only default mode network (DMN) showed significant positive contrasts).

**Figure 2 brainsci-11-01137-f002:**
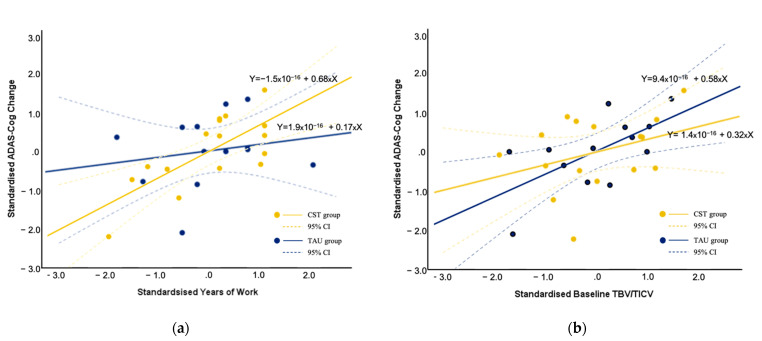
Relationship between (**a**) standardised years of work and (**b**) standardised baseline TBV/TICV with standardised ADAS-Cog changes controlled for age and gender. Positive values signify improvement on ADAS-Cog.

**Table 1 brainsci-11-01137-t001:** Baseline and follow-up demographic characteristics, cognitive reserve proxies, and cognition.

Mean (SD)/*n* (%)	Baseline			Follow-Up		
	CST (*n* = 16)	TAU (*n* = 13)	*t*/χ^2^	CST (*n* = 16)	TAU (*n* = 13)	*t*/χ^2^
Female, *n*	12 (75.0)	7 (53.85)	0.89	-	-	-
Age, years	80.65 (6.18)	79.92 (6.13)	−0.27	-	-	-
Duration of diagnosis, years	1.87 (1.64)	2.91 (1.72)	1.37	-	-	-
Education, years	3.41 (4.14)	6.15 (5.19)	1.61	-	-	-
Any formal education, *n*	7 (43.75)	12 (92.31)	8.29 **	-	-	-
Work, years	35.25 (15.02)	41.91 (12.57)	1.15	-	-	-
ADAS-Cog total	20.49 (6.76)	25.59 (9.82)	1.03	18.74 (6.48)	23.31 (8.71)	1.65
Memory & learning	16.39 (5.45)	18.44 (7.04)	0.75	14.91 (5.52)	18.46 (6.41)	1.62
Language	1.82 (1.94)	3.77 (4.19)	1.70	1.88 (1.58)	3.31 (3.40)	1.53
Praxis	1.94 (1.14)	1.38 (1.45)	−1.18	1.94 (1.85)	1.62 (1.04)	−0.57
CSDD	3.35 (5.27)	5.23 (10.09)	0.66	2.24 (2.54)	2.77 (3.35)	0.50
HCS	7.24 (5.75)	17.29 (9.86)	3.15 *	7.41 (5.09)	16.00 (8.43)	3.10 *
Grey matter, cm^3^	500.29 (38.89)	502.39 (54.21)	0.12	495.06 (39.01)	482.82 (50.18)	−0.73
White matter, cm^3^	392.53 (48.12)	387.54 (52.42)	−0.27	394.47 (49.22)	377.09 (52.89)	−0.89
TICV, cm^3^	1393.41 (137.39)	1439.31 (145.47)	0.88	1393.41 (134.40)	1429.64 (154.53)	0.66
GM/TICV	0.36 (0.02)	0.35 (0.02)	−1.31	0.36 (0.02)	0.34 (0.02)	−2.33 *
WM/TICV	0.28 (0.02)	0.27 (0.03)	−1.42	0.28 (0.02)	0.26 (0.03)	−2.04
TBV/TICV	0.64 (0.03)	0.62 (0.04)	−1.77	0.64 (0.02)	0.60 (0.04)	−2.81 *
DMN rs-FC, *r*	0.24 (0.10)	0.34 (0.21)	1.61	0.36 (0.19)	0.32 (0.13)	−0.66
CEN rs-FC, *r*	0.37 (0.19)	0.32 (0.12)	−0.66	0.36 (0.18)	0.31 (0.17)	−0.72
LAN rs-FC, *r*	0.27 (0.14)	0.28 (0.18)	0.29	0.24 (0.15)	0.27 (0.14)	−0.59
Time gap between behavioral and MRI assessments, days	42 (21.5)	23 (17.5)	2.99 **	64 (35.2)	14.5 (13.0)	4.87 ***

CST = Cognitive Stimulation Therapy; TAU = treatment as usual; ADAS-Cog = Alzheimer’s Disease Assessment Scale, Cognitive subscale; CSDD = Cornell Scale of Depression in Dementia; HCS = Holden Communication Scale; TICV = total intracranial volume; TBV = total brain volume; GM = grey matter; WM = white matter; DMN = default mode network; CEN = central executive network; LAN = language network; rs-FC = resting-state functional connectivity. * *p* < 0.05; ** *p* < 0.01; *** *p* < 0.001.

**Table 2 brainsci-11-01137-t002:** Within-subject changes in neuropsychological and MRI measurements after CST and TAU.

	CST (*n* = 16)		TAU (*n* = 13)	
	Mean (SD)	*t*	Mean (SD)	*t*
ADAS-Cog total *^a^*	+1.75 (4.92)	1.47	+0.28 (4.31)	0.23
Memory and learning	+1.81 (4.67)	1.60	−0.02 (3.41)	−0.02
Language	−0.05 (1.14)	−0.21	+0.46 (1.81)	0.92
Praxis	0.00 (1.58)	0	−0.23 (1.42)	−0.59
CSDD	+1.12 (3.97)	1.16	+2.47 (7.77)	1.14
HCS	−0.18 (6.55)	−0.83	+1.29 (2.14)	1.59
Grey matter, cm^3^	−5.24 (8.93)	−2.42 *	−10.73 (14.61)	−2.44 *
White matter, cm^3^	1.94 (7.95)	1.01	−2.45 (11.99)	−0.68
TICV, cm^3^	0.00 (18.36)	0	5.55 (10.80)	1.70
GM/TICV	−0.003 (0.01)	−1.98	−0.01 (0.01)	−2.45 *
WM/TICV	+0.001 (0.01)	0.78	−0.003 (0.01)	−1.31
TBV/TICV	+0.002 (0.01)	0.83	−0.01 (0.01)	4.39 **
DMN rs-FC, *r*	+0.12 (0.14)	3.31 *	−0.02 (0.21)	−0.32
CEN rs-FC, *r*	−0.01 (0.21)	−0.10	−0.01 (0.17)	−0.27
LAN rs-FC, *r*	−0.03 (0.16)	−0.70	−0.01 (0.19)	−0.25

CST = Cognitive Stimulation Therapy; TAU = treatment as usual; ADAS-Cog = Alzheimer’s Disease Assessment Scale, Cognitive subscale; CSDD = Cornell Scale of Depression in Dementia; HCS = Holden Communication Scale; TICV = total intracranial volume; TBV = total brain volume; GM = grey matter; WM = white matter; DMN = default mode network; CEN = central executive network; LAN = language network; rs-FC = resting-state functional connectivity. * *p* < 0.05; ** *p* < 0.01. *^a^* Positive values signify improvement.

**Table 3 brainsci-11-01137-t003:** Partial correlation between baseline network connectivity and cognitive changes controlling for age and gender, and Fisher’s Z transformation.

	ADAS-Cog Change *^a^*				
	CST Group (*n* = 16)		TAU Group (*n* = 13)		Comparison of Fisher’s-Z, z (*p*)
	*r*	Fisher’s-Z	*r*	Fisher’s-Z	
Work, years	0.75 **	0.97	0.20	0.20	1.83 (*p* = 0.06)
Baseline TBV/TICV	0.42	0.45	0.66 *	0.79	0.82 (*p* = 0.41)
Baseline DMN rs-FC, *r*	-0.41	-0.44	0.12	0.12	1.32 (*p* = 0.19)
Baseline CEN rs-FC, *r*	0.13	0.13	0.58	0.66	1.26 (*p* = 0.21)
Baseline LAN rs-FC, *r*	0.27	0.27	0.25	0.26	0.05 (*p* = 0.96)

TBV = total brain volume; TICV = total intracranial volume; DMN = default mode network; CEN = central executive network; LAN = language network; rs-FC = resting-state functional connectivity *^a^* Positive values signify improvement on ADAS-Cog. * *p* < 0.05; ** *p* < 0.01.

## Data Availability

No data are available. The ethical approval and participant consent for this study do not allow sharing of data beyond the research team.

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
