# Peer review of "Changes in Default Mode Network Connectivity in Resting-State fMRI in People with Mild Dementia Receiving Cognitive Stimulation Therapy"

_brainsci, 2021, doi:10.3390/brainsci11091137_

Round 1

Reviewer 1 Report

This article presents research investigating the effect of Group cognitive stimulation therapy (CST) in people with mild to moderate dementia using morphometric and functional MRI. Increased Default Mode Network connectivity was observed in participants in CST despite decrease grey matter volume. Whereas the topic investigated in this manuscript is interesting and well written, some methodological corrections need to be applied. Below are the specific comments regarding the manuscript.

  1. I would be helpful to visualize in a brain surface/volume the different ROIS corresponding to the 3 networks of interest: default mode network (DMN), central executive network (CEN) and language network (LAN). You could add this figure in supplementary material. There are many softwares that can generate figures with spheres. For example: https://www.nitrc.org/projects/bnv/
  2. Table 1 and Table 2: The rows Grey matter cm3 and White matter cm3 do not make a sense because this  vary with the head size of the participant. If you want to put a measure that you can compare between the 2 groups or correlate to the measures you need to control for the TICV. It will be more correct to put the mean and standard deviation of GM/TICV  of the participants (ratio gray matter to whole brain size).
  3. Table 1 and Table 2: What are the network connectivity betas? If I understood correctly the pearson correlation (r) is used for computing the roi to voxel connectivity and then a fisher transformation is applied (r to z). If this is the case the reported connectivity is not the beta is the correlation. The beta is usually used for the contribution of a variable when you perform a regression analysis, and I think this is no the case.
  4. Line 250: “Analyses using SPM based on connectivity matrix…” What connectivity matrix it should be explained in methods sections. Which regions are the nodes of the network, what type of analyses do you perform in the network? Link level statistic or somo network property of the connectivity matrix? Or instead of connectivity matrix you mean in the connectivity maps generated from the seeds?
  5. Figure 1: You should plot the T or Z statistics with a color palette that allows to appreciate regions with strongest statistical differences, not just the surviving voxels. To plot results in surface you have different options: coon toolbox provides tools to do this, also connectome workbench or the software recommended in point 1.
  6. Figure 2: It will be helpful to add the correlation values of each line in the figure.
  7. It is not clear for me what statistical analyses you performed with resting state data. Have you computed 3 statistical analysis one for each of the networks. Do you only observe results when comparing the DMN? No significant results for the rest? You should explain it better
  8. Figure 2: The correlation are corrected for age? Instead a pearson correlation a partial correlation to remove effect of age should be used.

Minor comments:

  1. Line 147: T1 images were used for structural morphometric analysis and for preprocessing of fMRI data.
  2. Line 165: Update sentence: isotropic Gaussian kernel to minimize between subject cortical variations of the gyrus.
  3. Line 189-195: This are preprocessing steps, you should move these sentences to the last part of preprocessing “In each participant… was not regressed.”

Author Response

This article presents research investigating the effect of Group cognitive stimulation therapy (CST) in people with mild to moderate dementia using morphometric and functional MRI. Increased Default Mode Network connectivity was observed in participants in CST despite decrease grey matter volume. Whereas the topic investigated in this manuscript is interesting and well written, some methodological corrections need to be applied. Below are the specific comments regarding the manuscript.

  1. It would be helpful to visualize in a brain surface/volume the different ROIS corresponding to the 3 networks of interest: default mode network (DMN), central executive network (CEN) and language network (LAN). You could add this figure in supplementary material. There are many softwares that can generate figures with spheres. For example: https://www.nitrc.org/projects/bnv/

Response: We thank the reviewer for the suggestion, we added a brain surface map with different ROIs corresponding to the three networks of interest (Supplementary Figure 1).

  1. Table 1 and Table 2: The rows Grey matter cm3 and White matter cm3 do not make sense because this vary with the head size of the participant. If you want to put a measure that you can compare between the 2 groups or correlate to the measures you need to control for the TICV. It will be more correct to put the mean and standard deviation of GM/TICV of the participants (ratio gray matter to whole brain size).

Response: We thank the reviewer for the comment. We have now revised Table 1 and Table 2 by reporting GM/TICV and WM/TICV instead of absolute values. We have also removed text describing absolute volume reduction in the manuscript.

  1. Table 1 and Table 2: What are the network connectivity betas? If I understood correctly the pearson correlation (r) is used for computing the roi to voxel connectivity and then a fisher transformation is applied (r to z). If this is the case the reported connectivity is not the beta is the correlation. The beta is usually used for the contribution of a variable when you perform a regression analysis, and I think this is not the case.

Response: We thank reviewer for spotting this. We had indeed opted for the correlation (r) instead regression (beta) option for first-level connectivity analyses using the Conn Toolbox. We have now corrected this in all tables. 

  1. Line 250: “Analyses using SPM based on connectivity matrix…” What connectivity matrix it should be explained in methods sections. Which regions are the nodes of the network, what type of analyses do you perform in the network? Link level statistic or somo network property of the connectivity matrix? Or instead of connectivity matrix you mean in the connectivity maps generated from the seeds?

Response: We apologise for the lack of clarity in our writing. We meant to say seed-based functional connectivity maps. For clarification, we have revised this paragraph.

  1. Figure 1: You should plot the T or Z statistics with a color palette that allows to appreciate regions with strongest statistical differences, not just the surviving voxels. To plot results in surface you have different options: coon toolbox provides tools to do this, also connectome workbench or the software recommended in point 1.

Response: We thank the reviewer for this suggestion, we have updated Figure 1 (surface display) with color bar and relevant statistics, we also updated the figure caption.

  1. Figure 2: It will be helpful to add the correlation values of each line in the figure.

Response: Take into consideration both comment 6 and comment 8, we have performed linear regression to control for the effects of age and gender, and used standardised ADAS-Cog change score residual to make the scatterplot. The coefficients of each line as well as the intercepts are now included in Figure 2.

  1. It is not clear for me what statistical analyses you performed with resting state data. Have you computed 3 statistical analysis one for each of the networks. Do you only observe results when comparing the DMN? No significant results for the rest? You should explain it better

Response: We thank the reviewer for pointing this out, and updated the text about statistical analyses. We computed three analyses for each of the networks, and significant results were observed only with DMN. The results are updated.

  1. Figure 2: The correlation are corrected for age? Instead a pearson correlation a partial correlation to remove effect of age should be used.

Response: We performed partial correlation to remove effects of age and gender (since gender was correlated with brain measures in both groups, Table S1). The partial correlation results are presented in lines 298-302, also summarised in Table 3.

Minor comments:

  1. Line 147: T1 images were used for structural morphometric analysis and for preprocessing of fMRI data.

Response: Thank you. This is now corrected.

  1. Line 165: Update sentence: isotropic Gaussian kernel to minimize between subject cortical variations of the gyrus.

Response: Thank you. This is updated.

  1. Line 189-195: This are preprocessing steps, you should move these sentences to the last part of preprocessing “In each participant… was not regressed.

Response: We thank the reviewer for pointing this out. This paragraph actually describes the denoising process, we apologies for not articulating it clearly. We have now amended this section and placed it to the last part of preprocessing.

Reviewer 2 Report

The authors examined the changes in both structural morphometry and functional connectivity of patients with mild dementia who received CST as compared to control patient group. The authors are interested in the three important networks – DMN, CEN and LAN. And they found significant changes in the DMN connectivity. This is a very interesting study. However, I still have some questions that needs to be addressed.

Abstract

1) Line 22: Besides resting-state fMRI, VBM should also be mentioned in the sentence.

2) Line 23: It might be more rigorous if the authors could let the readers know that the dementia controls received treatment as usual (TAU). Otherwise, it sounds like the controls were not receiving any treatments.

3) Lines 25-26: The sentence that described the results is not very clear.

Introduction

1) Lines 97-98: Reference 28 is a book. Could the authors provide more details about this study? In particular, the VBM study found higher TIV at baseline is associated with improvements in general cognition at post-CST. Does it mean the correlation has been performed between TIV at baseline and the differences in general cognition computed as (post-CST vs. baseline)? And what was the measure for general cognition? IQ test?

Materials and Methods

1) The numbers of participants (each group) who have been excluded because of excessive head motion, incompletion of follow-up scan, etc as well as the final sample size should be reported in the section 2.1, rather than in the other sections of Materials and Methods and the first paragraph in the part of Results.

2) Line 121: an extra comma

3) Lines 141-142: The MRI scan time is not described clearly. When was the second scan? On average within 42 days (after the behavioral assessments?) at T1? And why the scans were arranged right after the behavioral assessments, or at least with a shorter time gap? Will the gap between scan and behavioral assessment be a confounding factor? Could the author show the mean and SD of the days between scan and behavioral assessments for each group?

4) Lines 167-168, how many datasets have been discarded?

5) Although the datasets with excessive head movement have been discarded, head movement may also have influences on the data for the remaining subjects. In the preprocessing, do the authors consider scrubbing the data for each subject based on FD and DVARS, as typically done in the resting state fMRI analysis?

Results

1) Lines 246-253: Have the covariates of age, gender and education been put into the analyses?

2) Figure 1: the figure caption is not clear. And there is no color bar for the T values of functional connectivity change map.

3) Can the authors add 95% CI to the regression lines in Figure 2?

4) The description of Figure 2 in the main text is not clear. I cannot find what is the r between years of work and change in ADAS-Cog Total for the control group?

5) The correlations in CST vs control groups can be compared statistically using slope tests or Fisher’s Z transformation, which may help boost the conclusion. For instance, in Figure 2a, for the CST group, the correlation between change in ADAS-Cog Total and Years of Work was 0.45, and for the control group, the correlation was a negative value (although not shown in the text). Whether the correlation (r = 0.45) for the CST group is significantly different from the correlation (r = negative value) for the control group can be tested with slope tests/Fisher’s Z transformation. And this can be applied to all comparisons of correlations between groups, such as for Table 3. For instance, -0.14 vs 0.41, 0.15 vs 0.68, and 0.28 vs 0.12, the authors can report the p values of the comparisons of r.

Author Response

Abstract

1) Line 22: Besides resting-state fMRI, VBM should also be mentioned in the sentence.

Response: Thank you. This has now been corrected.

2) Line 23: It might be more rigorous if the authors could let the readers know that the dementia controls received treatment as usual (TAU). Otherwise, it sounds like the controls were not receiving any treatments.

Response: We thank the reviewer for pointing this out, the abstract is now updated.

3) Lines 25-26: The sentence that described the results is not very clear.

Response: We updated the sentences about structural and functional MRI results.

Introduction

1) Lines 97-98: Reference 28 is a book. Could the authors provide more details about this study? In particular, the VBM study found higher TIV at baseline is associated with improvements in general cognition at post-CST. Does it mean the correlation has been performed between TIV at baseline and the differences in general cognition computed as (post-CST vs. baseline)? And what was the measure for general cognition? IQ test?

Response: Thank you. In this previous study (findings reported in a book chapter), correlation analyses were performed between normalised grey matter volume and white matter volume at baseline (i.e., GM/TIV and WM/TIV ratio) with improvements in ADAS-Cog score (post-CST vs. baseline), the correlation score were r=0.47 and 0.46, respectively, p<0.05. Cognition was measured by ADAS-Cog. We have now supplemented more details about this study.

Materials and Methods

1) The numbers of participants (each group) who have been excluded because of excessive head motion, incompletion of follow-up scan, etc as well as the final sample size should be reported in the section 2.1, rather than in the other sections of Materials and Methods and the first paragraph in the part of Results.

Response: We have now moved the paragraph in the first part of the results and the related sentences in the materials and methods to section 2.1.

2) Line 121: an extra comma

Response: Thank you. This has now been corrected.

3) Lines 141-142: The MRI scan time is not described clearly. When was the second scan? On average within 42 days (after the behavioral assessments?) at T1? And why the scans were arranged right after the behavioral assessments, or at least with a shorter time gap? Will the gap between scan and behavioral assessment be a confounding factor? Could the author show the mean and SD of the days between scan and behavioral assessments for each group?

Response: We thank the reviewer for pointing this out. We have now report the time gap between behavioral assessment and MRI scan at T0 and T1 for both groups in Table 1. The rationale for arranging the scans shortly after behavioural assessments is to minimise the risk of change in condition, which could be common in older population with dementia (e.g., fall), and the potentially confounding effects from other interventions they may receive as the participants are known to existing health and social care services. Therefore, in theory the gap is a potential confounding factor. However based on our bivariate correlation analysis the time intervals were not correlated with cognitive change in either group (see Supplementary Table 1), and therefore time interval was not included as a control variable in the analyses.

4) Lines 167-168, how many datasets have been discarded?

Response: No dataset was discarded because of excessive head motion, we updated it now in lines 174-176.

5) Although the datasets with excessive head movement have been discarded, head movement may also have influences on the data for the remaining subjects. In the preprocessing, do the authors consider scrubbing the data for each subject based on FD and DVARS, as typically done in the resting state fMRI analysis?

Response: We thank the reviewer for pointing this out, a paragraph detailing the procedure of denoising the data after pre-processing is now included. The CONN toolbox default denoising pipeline includes linear regression of scrubbing parameters. For each identified outlier scan during the outlier identification preprocessing step was used as potential confounding effects to remove any influence of these outlier scans on the BOLD signal.  

Results

1) Lines 246-253: Have the covariates of age, gender and education been put into the analyses?

Response: In the paired-sample t-tests, the covariates were not adjusted. Years of education in this low-education sample was not related to ADAS-Cog score at baseline or follow-up, or its change; so, we ran repeated measures ANCOVA with age and gender as covariates, and reported the results. No significant change was found after adjusting for these covariates in separate group analyses. However, the mixed ANCOVA analyses between two groups while adjusting for covariates found significant interaction between timepoints and group on TBV/TICV and marginal interaction on rs-FC of DMN. The updated results are highlighted.

2) Figure 1: the figure caption is not clear. And there is no color bar for the T values of functional connectivity change map.

Response: We thank the reviewer for the suggestion, and we updated Figure 1 with color bar and relevant statistics, and Figure caption was updated as well.

3) Can the authors add 95% CI to the regression lines in Figure 2?

Response: We thank the reviewer for pointing this out, 95% CI are now included to the regression lines.

4) The description of Figure 2 in the main text is not clear. I cannot find what is the r between years of work and change in ADAS-Cog Total for the control group?

Response: Years of work and change in ADAS-Cog total for the control group (TAU) were not significantly correlated (r=-0.08), details are presented in Table S1. The regression coefficient in Figure 2a was positive (B=0.17) because age and gender were controlled. Raw correlation coefficients between different key variables are reported in supplementary Table 1.

5) The correlations in CST vs control groups can be compared statistically using slope tests or Fisher’s Z transformation, which may help boost the conclusion. For instance, in Figure 2a, for the CST group, the correlation between change in ADAS-Cog Total and Years of Work was 0.45, and for the control group, the correlation was a negative value (although not shown in the text). Whether the correlation (r = 0.45) for the CST group is significantly different from the correlation (r = negative value) for the control group can be tested with slope tests/Fisher’s Z transformation. And this can be applied to all comparisons of correlations between groups, such as for Table 3. For instance, -0.14 vs 0.41, 0.15 vs 0.68, and 0.28 vs 0.12, the authors can report the p values of the comparisons of r.

Response: We thank the reviewer for this very helpful suggestion. We performed Fisher’s Z transformation of the partial correlation coefficients after controlling for age and gender, and the results are reported in Table 3 now. There was marginal difference between two groups in correlation between work years and cognitive improvement (p=0.06), and all other differences were non-significant.